# LEARNING TO DIAGNOSE FROM SCRATCH BY EXPLOITING DEPENDENCIES AMONG LABELS

## ABSTRACT

The field of medical diagnostics contains a wealth of challenges which closely resemble classical machine learning problems; practical constraints, however, complicate the translation of these endpoints naively into classical architectures. Many tasks in radiology, for example, are largely problems of multi-label classification wherein medical images are interpreted to indicate multiple present or suspected pathologies. Clinical settings drive the necessity for high accuracy simultaneously across a multitude of pathological outcomes and greatly limit the utility of tools which consider only a subset. This issue is exacerbated by a general scarcity of training data and maximizes the need to extract clinically relevant features from available samples – ideally without the use of pre-trained models which may carry forward undesirable biases from tangentially related tasks. We present and evaluate a partial solution to these constraints in using LSTMs to leverage interdependencies among target labels in predicting 14 pathologic patterns from chest x-rays and establish state of the art results on the largest publicly available chest x-ray dataset from the NIH without pre-training. Furthermore, we propose and discuss alternative evaluation metrics and their relevance in clinical practice.

## 1 INTRODUCTION

Medical diagnostics have increasingly become a more interesting and viable endpoint for machine learning. A general scarcity of publicly available medical data, however, inhibits its rapid development. Pre-training on tangentially related datasets such as ImageNet (Deng et al., 2009) has been shown to help in circumstances where training data is limited, but may introduce unintended biases which are undesirable in a clinical setting. Furthermore, most clinical settings will drive a need for models which can accurately predict a large number of diagnostic outcomes. This essentially turns many medical problems into multi-label classification with a large number of targets, many of which may be subtle or poorly defined and are likely to be inconsistently labeled. In addition, unlike the traditional multi-label setting, predicting the absence of each label is as important as predicting its presence in order to minimize the possibility of misdiagnosis. Each of these challenges drive a need for architectures which consider clinical context to make the most of the data available.

Chest x-rays are the most common type of radiology exam in the world and a particularly challenging example of multi-label classification in medical diagnostics. Making up nearly 45% of all radiological studies, the chest x-ray has achieved global ubiquity as a low-cost screening tool for a wealth of pathologies including lung cancer, tuberculosis, and pneumonia. Each scan can contain dozens of patterns corresponding to hundreds of potential pathologies and can thus be difficult to interpret, suffering from high disagreement rates between radiologists and often resulting in unnecessary follow-up procedures. Complex interactions between abnormal patterns frequently have significant clinical meaning that provides radiologists with additional context. For example, a study labeled to indicate the presence of cardiomegaly (enlargement of the cardiac silhouette) is more likely to additionally have pulmonary edema (abnormal fluid in the extravascular tissue of the lung) as the former may suggest left ventricular failure which often causes the latter. The presence of edema further predicates the possible presence of both consolidation (air space opacification) and a pleural effusion (abnormal fluid in the pleural space). Training a model to recognize the potential for these interdependencies could enable better prediction of pathologic outcomes across all categories while maximizing the data utilization and its statistical efficiency.

Among the aforementioned challenges, this work firstly addresses the problem of predicting multiple labels simultaneously while taking into account their conditional dependencies during both the training and the inference. Similar problems have been raised and analyzed in the work of Wang et al. (2016); Chen et al. (2017) with the application of image tagging, both outside the medical context. The work of Shin et al. (2016); Wang et al. (2017) for chest x-ray annotations are closest to ours. All of them utilize out-of-the-box decoders based on recurrent neural networks (RNNs) to sequentially predict the labels. Such a naive adoption of RNNs is problematic and often fails to attend to peculiarities of the medical problem in their design, which we elaborate on in Section 2.3 and Section 3.3.1.

In addition, we hypothesize that the need for pre-training may be safely removed when there are sufficient medical data available. To verify this, all our models are trained from scratch, without using any extra data from other domains. We directly compare our results with those of Wang et al. (2017) that are pre-trained on ImageNet. Furthermore, to address the issue of clinical interpretability, we juxtapose a collection of alternative metrics along with those traditionally used in machine learning, all of which are reported in our benchmark.

## 1.1 MAIN CONTRIBUTIONS

This work brings state-of-the-art machine learning models to bear on the problem of medical diagnosis with the hope that this will lead to better patient outcomes. We have advanced the existing research in three orthogonal directions:

- This work experimentally verifies that without pre-training, a carefully designed baseline model that ignores the label dependencies is able to outperform the pre-trained state-of-the-art by a large margin.

- A collection of metrics is investigated for the purpose of establishing clinically relevant and interpretable benchmarks for automatic chest x-ray diagnosis.

- We propose to explicitly exploit the conditional dependencies among abnormality labels for better diagnostic results. Existing RNNs are purposely modified to accomplish such a goal. The results on the proposed metrics consistently indicate their superiority over models that do not consider interdependencies.

## 2 RELATED WORK

### 2.1 NEURAL NETWORKS IN MEDICAL IMAGING

The present work is part of a recent effort to harness advances in Artificial Intelligence and machine learning to improve computer-assisted diagnosis in medicine. Over the past decades, the volume of clinical data in machine-readable form has grown, particularly in medical imaging. While previous generations of algorithms struggled to make effective use of this high-dimensional data, modern neural networks have excelled at such tasks. Having demonstrated their superiority in solving difficult problems involving natural images and videos, recent surveys from Litjens et al. (2017); Shen et al. (2017); Qayyum et al. (2017) suggest that they are rapidly becoming the "de facto" standard for classification, detection, and segmentation tasks with input modalities such as CT, MRI, x-ray, and ultrasound. As further evidence, models based on neural networks dominate the leaderboard in most medical imaging challenges [1,2].

Most successful applications of neural networks to medical images rely to a large extent on convolutional neural networks (ConvNets), which were first proposed in LeCun et al. (1998). This comes as no surprise since ConvNets are the basis of the top performing models for natural image understanding. For abnormality detection and segmentation, the most popular variants are UNets from Ronneberger et al. (2015) and VNets from Milletari et al. (2016), both built on the idea of fully convolutional neural networks introduced in Long et al. (2015). For classification, representative examples of neural network-based models from the medical literature include: Esteva et al. (2017) for skin cancer classification, Gulshan et al. (2016) for diabetic retinopathy, Lakhani & Sundaram

---

[1] https://grand-challenge.org
[2] https://www.kaggle.com/c/data-science-bowl-2017

(2017) for pulmonary tuberculosis detection in x-rays, and Huang et al. (2017b) for lung cancer diagnosis with chest CTs. All of the examples above employed 2D or 3D ConvNets and all of them provably achieved near-human level performance in their particular setup. Our model employs a 2D ConvNet as an image encoder to process chest x-rays.

## 2.2 MULTI-LABEL CLASSIFICATION

Given a finite set of possible labels, the multi-label classification problem is to associate each instance with a subset of those labels. Being relevant to applications in many domains, a variety of models have been proposed in the literature. The simplest approach, known as binary relevance, is to break the multi-label classification problem into independent binary classification problems, one for each label. A recent example from the medical literature is Wang et al. (2017). The appeal of binary relevance is its simplicity and the fact that it allows one to take advantage of a rich body of work on binary classification. However, it suffers from a potentially serious drawback: the assumption that the labels are independent. For many applications, such as the medical diagnostic application motivating this work, there are significant dependencies between labels that must be modeled appropriately in order to maximize the performance of the classifier.

Researchers have sought to model inter-label dependencies by making predictions over the label power set (e.g. Tsoumakas & Vlahavas (2007) and Read et al. (2008)), by training classifiers with loss functions that implicitly represent dependencies (e.g. Li et al. (2017)), and by using a sequence of single-label classifiers, each of which receives a subset of the previous predictions along with the instance to be classified (e.g. Dembczyński et al. (2012)). The later approach is equivalent to factoring the joint distribution of labels using a product of conditional distributions. Recent research has favored recurrent neural networks (RNNs), which rely on their state variables to encode the relevant information from the previous predictions (e.g. Wang et al. (2016) and Chen et al. (2017)). The present work falls into this category.

## 2.3 KEY DIFFERENCES

To detect and classify abnormalities in chest x-ray images, we propose using 2D ConvNets as encoders and decoders based on recurrent neural networks (RNNs). Recently, Lipton et al. (2016) proposed an RNN-based model for abnormality classification that, based on the title of their paper, bears much resemblance to ours. However, in their work the RNN is used to process the inputs rather than the outputs, which fails to capture dependencies between labels; something we set out to explicitly address. They also deal exclusively with time series data rather than high-resolution images.

The work of Shin et al. (2016) also addresses the problem of chest x-ray annotation. They built a cascaded three-stage model using 2D ConvNets and RNNs to sequentially annotate both the abnormalities and their attributes (such as location and severity). Their RNN decoder resembles ours in its functionality, but differs in the way the sequence of abnormalities are predicted. In each RNN step, their model predicts one of $T$ abnormalities with softmax, and stops when reaching a predefined upper limit of total number of steps (5 is used in theirs). Instead, our model predicts the presence or absence of $t$-th abnormality with sigmoid at time step $t$ and the total number of steps is the number of abnormalities. The choice of such a design is inspired by Neural Autoregressive Density Estimators (NADEs) of Larochelle & Murray (2011). Being able to predict the absence of an abnormality and feed to the next step, which is not possible with softmax and argmax, is preferable in the clinical setting to avoid any per-class overcall and false alarm. In addition, the absence of a certain abnormality may be a strong indication of the presence or absence of others. Beyond having a distinct approach to decoding, their model was trained on the OpenI[3] dataset with 7000 images, which is smaller and less representative than the dataset that we used (see below). In addition, we propose a different set of metrics to use in place of BLEU (Papineni et al., 2002), commonly used in machine translation, for better clinical interpretation.

In the non-medical setting, Wang et al. (2016) proposed a similar ConvNet–RNN architecture. Their choice of using an RNN decoder was also motivated by the desire to model label dependencies. However, they perform training and inference in the manner of Shin et al. (2016). Another example

---

[3]https://openi.nlm.nih.gov

of this combination of application, architecture, and inference comes from Chen et al. (2017) whose work focused on eliminating the need to use a pre-defined label order for training. We show in the experiments that ordering does not seem to impose as a significant constraint when models are sufficiently trained.

Finally, Wang et al. (2017) proposed a 2D ConvNet for classifying abnormalities in chest x-ray images. However, they used a simple binary relevance approach to predict the labels. As we mentioned earlier, there is strong clinical evidence to suggest that labels do in fact exhibit dependencies that we attempt to model. They also presented the largest public x-ray dataset to date ("ChestX-ray8"). Due to its careful curation and large volume, such a collection is a more realistic retrospective clinical study than OpenI and therefore better suited to developing and benchmarking models. Consequently, we use "ChestX-ray8" to train and evaluate our model. And it should be noted that unlike Wang et al. (2017), we train our models from scratch to ensure that the image encoding best captures the features of x-ray images as opposed to natural images.

## 3 MODELS

The following notations are used throughout the paper. Denote $\mathbf{x}$ as an input image, and $\mathbf{x} \in \mathcal{R}^{w \times h \times c}$ where $w$, $h$ and $c$ represent width, height, and channel. Denote $\mathbf{y}$ as a binary vector of dimensionality $T$, the total number of abnormalities. We used superscripts to indicate a specific dimensionality. Thus, given a specific abnormality $t$, $\mathbf{y}^t = 0$ indicates its absence and $\mathbf{y}^t = 1$ its presence. We use subscripts to index a particular example, for instance, $\{\mathbf{x}_i, \mathbf{y}_i\}$ is the $i$-th example. In addition, $\theta$ denotes the union of parameters in a model. We also use $\mathbf{m}$ to represent a vector with each element $\mathbf{m}^t$ as the mean of a Bernoulli distribution.

### 3.1 DENSELY CONNECTED IMAGE ENCODER

A recent variant of Convolutional Neural Network (ConvNet) is proposed in Huang et al. (2017a), dubbed as Densely Connected Networks (DenseNet). As a direct extension of Deep Residual Networks (He et al., 2016) and Highway Networks (Srivastava et al., 2015), the key idea behind DenseNet is to establish shortcut connections from all pairs of layers at different depth of a very deep neural network. It has been argued in Huang et al. (2017a) that, as the result of the extensive and explicit feature reuse in DenseNets, they are both computationally and statistically more efficient. This property is particularly desirable in dealing with medical imaging problems where the number of training examples are usually limited and overfitting tends to prevail in models with more than tens of millions of parameters.

We therefore propose a model based on the design of DenseNets while taking into account the peculiarity of medical problems at hand. Firstly, the inputs of the model are of much higher resolutions. Lower resolution, typically with $256 \times 256$, may be sufficient in dealing with problems related to natural images, photos and videos, a higher resolution, however, is often necessary to faithfully represent regions in images that are small and localized. Secondly, the proposed model is much smaller in network depth. While there is ample evidence suggesting the use of hundreds of layers, such models typically require hundreds of thousands to millions of examples to train. Large models are prone to overfitting with one tenth the training data. Figure 1 highlights such a design.

### 3.2 INDEPENDENT PREDICTION OF LABELS

Ignoring the nature of conditional dependencies among the indicators, $\mathbf{y}^t$, one could establish the following probabilistic model:

$$P(\mathbf{y}|\mathbf{x}) = \prod_{t=1}^{T} P(\mathbf{y}^t|\mathbf{x}). \tag{1}$$

Equ (1) assumes that knowing one label does not provide any additional information about any other label. Therefore, in principle, one could build a separate model for each $\mathbf{y}^t$ which do not share any parameters. However, it is common in the majority of multi-class settings to permit a certain degree of parameter sharing among individual classifiers, which encourages the learned features to be reused among them. Furthermore, sharing alleviates the effect of overfitting as the example-parameter ratio is much higher.

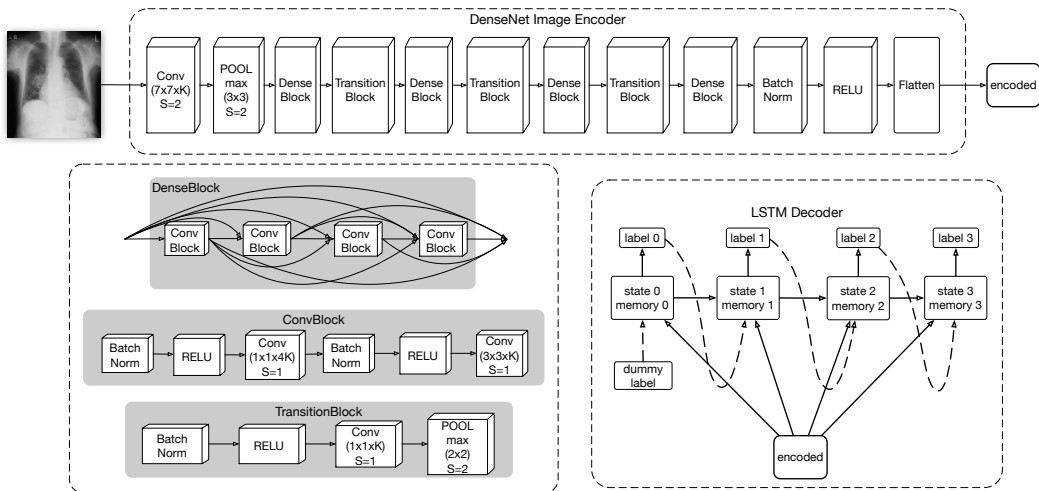

Figure 1: The input image is encoded by a densely connected convolutional neural network (top). Similar to DenseNets from Huang et al. (2017a), our variant consists of DenseBlocks and TransitionBlocks. Within each DenseBlock, there are several ConvBlocks. The resulting encoded representation of the input is a vector that captures the higher-order semantics that are useful for the decoding task. $K$ is the growth rate in Huang et al. (2017a), $S$ is the stride. We also include the filter and pooling dimensionality when applicable. Unlike a DenseNet that has 16 to 32 ConvBlock within a DenseBlock, our model uses 4 in order to keep the total number of parameters small. Our proposed RNN decoder is illustrated on the bottom right.

### 3.2.1 TRAINING

During training, the model optimizes the following Maximum Log-likelihood Estimate (MLE) criteria:

$$\mathcal{L}_{\alpha} = \arg\max_{\theta} \sum_{t=1}^{T} \log P(\mathbf{y}^t|\mathbf{x}, \theta) \tag{2}$$

where $P(\mathbf{y}^t|\mathbf{x}, \theta)$ is a Bernoulli distribution with its mean $\mathbf{m}^t$ parameterized by the model. In particular, $\mathbf{m}^t = \text{sigmoid}(f(\mathbf{x}, \theta))$.

### 3.2.2 INFERENCE

As labels are considered independent and during the inference, a binary label is generated for each factor independently with $\mathbf{y}^{t^*} = \arg\max P(\mathbf{y}^t|\mathbf{x}, \theta)$. This is equivalent to setting the classification threshold to 0.5.

### 3.3 EXPLOITING HIGHER-ORDER DEPENDENCIES AMONG LABELS

As discussed in length in Section 1, it is hardly true that abnormalities are independent from each other. Hence the assumption made by Equ (1) is undoubtably too restrictive. In order to treat the multi-label problem in its full generality, we can begin with the following factorization, which makes no assumption of independence:

$$P(\mathbf{y}|\mathbf{x}) = P(\mathbf{y}^0|\mathbf{x})P(\mathbf{y}^1|\mathbf{y}^0, \mathbf{x}) \dots P(\mathbf{y}^T|\mathbf{y}^0, \dots, \mathbf{y}^{T-1}, \mathbf{x}). \tag{3}$$

Here, the statistical dependencies among the indicators, $\mathbf{y}^t$, are explicitly modeled within each factor so the absence or the presence of a particular abnormality may suggest the absence or presence of others.

The factorization in Equ (3) has been the central study of many recent models. Bengio & Bengio (2000) proposed the first neural network based model, refined by Larochelle & Murray (2011) and Gregor et al. (2014), all of which used the model in the context of unsupervised learning in small discrete data or small image patches. Recently Sutskever et al. (2014); Cho et al. (2014) popularized

the so-called "sequence-to-sequence" model where a Recurrent Neural Network (RNN) decoder models precisely the same joint distribution while conditioned on the output of an encoder. Compared with the previous work, RNNs provide a more general framework to model Equ (3) and an unmatched proficiency in capturing long term dependencies when $K$ is large.

We therefore adopt the Long-short Term Memory Networks (LSTM) (Hochreiter & Schmidhuber, 1997) and treat the multi-label classification as sequence prediction with a fixed length. The formulation of our LSTM is particularly similar to those used in image and video captioning (Xu et al., 2015; Yao et al., 2015), but without the use of an attention mechanism and without the need of learning when to stop.

Given an input $\mathbf{x}$, the same DenseNet-based encoder of Section 3.2 is applied to produce a lower dimensional vector representation of it with

$$\mathbf{x}_{\text{enc}} = f_{\text{enc}}(\mathbf{x}) \tag{4}$$

For the decoder, $\mathbf{x}_{\text{enc}}$ is used to initialize both the states and memory of an LSTM with

$$\mathbf{h}_0 = f_{h_0}(\mathbf{x}_{\text{enc}}) \qquad\qquad \mathbf{c}_0 = f_{c_0}(\mathbf{x}_{\text{enc}}) \tag{5}$$

where $f_{h_0}$ and $f_{c_0}$ are standard feedforward neural networks with one hidden layer. With $\mathbf{h}_0$ and $\mathbf{c}_0$, the LSTM decoder is parameterized as

$$\mathbf{g}_i^t = \text{sigmoid}(\mathbf{x}_{\text{enc}}\mathbf{W}_i + \mathbf{h}^{t-1}\mathbf{U}_i + \mathbf{y}^{t-1}\mathbf{V}_i + \mathbf{b}_i) \tag{6}$$

$$\mathbf{g}_o^t = \text{sigmoid}(\mathbf{x}_{\text{enc}}\mathbf{W}_o + \mathbf{h}^{t-1}\mathbf{U}_o + \mathbf{y}^{t-1}\mathbf{V}_o + \mathbf{b}_o) \tag{7}$$

$$\mathbf{g}_f^t = \text{sigmoid}(\mathbf{x}_{\text{enc}}\mathbf{W}_f + \mathbf{h}^{t-1}\mathbf{U}_f + \mathbf{y}^{t-1}\mathbf{V}_f + \mathbf{b}_f) \tag{8}$$

$$\mathbf{g}_g^t = \mathbf{x}_{\text{enc}}\mathbf{W}_g + \mathbf{h}^{t-1}\mathbf{U}_g + \mathbf{y}^{t-1}\mathbf{V}_g + \mathbf{b}_g \tag{9}$$

$$\mathbf{c}^t = \mathbf{g}_f^t \odot \mathbf{c}^{t-1} + \mathbf{g}_i^t \odot \tanh(\mathbf{g}_g^t) \tag{10}$$

$$\mathbf{h}^t = \mathbf{g}_o^t \odot \tanh(\mathbf{c}^t) \tag{11}$$

$$\mathbf{m}^t = \text{sigmoid}(\mathbf{h}^t\mathbf{q}^T + b_l) \tag{12}$$

where model parameters consist of three matrices $\mathbf{W}$s, $\mathbf{U}$s, $\mathbf{V}$s, vectors $\mathbf{b}$s and a scalar $b_l$. $\mathbf{y}$ is a vector code of the ground truth labels that respects a fixed ordering, with each element being either 0 or 1. All the vectors, including $\mathbf{h}$s, $\mathbf{g}$s, $\mathbf{c}$s, $\mathbf{b}$s, $\mathbf{q}$ and $\mathbf{y}$ are row vectors such that the vector-matrix multiplication makes sense. $\odot$ denotes the element-wise multiplication. Both sigmoid and tanh are element-wise nonlinearities. For brevity, we summarize one step of decoder computation as

$$\mathbf{m}^t = f_{\text{dec}}(\mathbf{x}_{\text{enc}}, \mathbf{y}^{t-1}, \mathbf{h}^{t-1}) \tag{13}$$

where the decoder LSTM computes sequentially the mean of a Bernoulli distribution. With Equ (3), each of its factor may be rewritten as

$$P(\mathbf{y}^t|\mathbf{y}^0, \ldots, \mathbf{y}^{t-1}, \mathbf{x}) = P(\mathbf{y}^t = 1)^{\mathbf{m}^t} P(\mathbf{y}^t = 0)^{(1-\mathbf{m}^t)} \tag{14}$$

### 3.3.1 THE DESIGN CHOICE OF SIGMOID

The choice of using sigmoid to predict $\mathbf{y}_t$ is by design. Standard sequence-to-sequence models often use softmax to predict one out of T classes and thus need to learn explicitly an "end-of-sequence" class. This is not desirable in our context due to the sparseness of the labels, resulting in the learned decoder being strongly biased towards predicting "end-of-sequence" while missing infrequently appearing abnormalities. Secondly, during the inference of the softmax based RNN decoder, the prediction at the current step is largely based on the presence of abnormalities at all previous steps due to the use of argmax. However, in the medical setting, the absence of previously predicted abnormalities may also be important. Sigmoid conveniently addresses these issues by explicitly predicting 0 or 1 at each step and it does not require the model to learn when to stop; the decoder always runs for the same number of steps as the total number of classes. Figure 1 contains the overall architecture of the decoder.

### 3.3.2 TRAINING

During training, the model optimizes

$$\mathcal{L}_\beta = \arg\max_\theta \sum_{t=1}^{T} \log P(\mathbf{y}^k|\mathbf{y}^0, \ldots, \mathbf{y}^{t-1}, \mathbf{x}, \theta) \tag{15}$$

Compared with Equ (1), the difference is the explicit dependencies among $\mathbf{y}^t$s. One may also notice that such a factorization is not unique – in fact, there exist $T!$ different orderings. Although mathematically equivalent, in practice, some of the orderings may result in a model that is easier to train. We investigate in Section 4 the impact of such decisions with two distinct orderings.

### 3.3.3 INFERENCE

The inference of such a model is unfortunately intractable as $\mathbf{y}^* = \arg\max_{\mathbf{y}} P(\mathbf{y}^0, \ldots, \mathbf{y}^T | \mathbf{x})$. Beam search (Sutskever et al., 2014) is often used as an approximation. We have found in practice that greedy search, which is equivalent to beam search with size 1, results in similar performance due to the binary sampling nature of each $\mathbf{y}^t$, and use it throughout the experiments. It is equivalent to setting 0.5 as the discretization threshold on each of the factors.

## 4 EXPERIMENTS

### 4.1 DATASET

To verify the efficacy of the proposed models in medical diagnosis, we conduct experiments on the dataset introduced in Wang et al. (2017). It is to-date the largest collection of chest x-rays that is publicly available. It contains in total 112,120 frontal-view chest x-rays each of which is associated with the absence or presence of 14 abnormalities. The dataset is originally released in PNG format, with each image rescaled to $1024 \times 1024$.

As there is no standard split for this dataset, we follow the guideline in Wang et al. (2017) to randomly split the entire dataset into 70% for training, 10% for validation and 20% for training. The authors of Wang et al. (2017) noticed insignificant performance difference with different random splits, as confirmed in our experiments by the observation that the performance on validation and test sets are consistent with each other.

### 4.2 PERFORMANCE METRICS

As the dataset is relatively new, the complete set of metrics have yet to be established. In this work, the following metrics are considered, and their advantage and drawbacks outlined below.

1. **Negative log-probability of the test set (NLL)**. This metric has a direct and intuitive probabilistic interpretation: The lower the NLL, the more likely the ground truth label. However, it is difficult to associate it with a clinical interpretation. Indeed, it does not directly reflect how accurate the model is at diagnosing cases with or without particular abnormalities.

2. **Area under the ROC curves (AUC)**. This is the reported metric of Wang et al. (2017) and it is widely used in modern biostatistics to measure collectively the rate of true detections and false alarms. In particular, we define 4 quantities: (1) true positive as TP: model predicts 1 with ground truth 1. (2) true negative as TN: model predicts 0 with ground truth 0. (3) false positive as FP: model predicts 1 with ground truth 0. (4) false negative as FN: model predicts 0 with ground truth 1. **Sensitivity (or recall)** is computed as $\mathrm{TP}/(\mathrm{TP} + \mathrm{FN})$ that measures the success of identifying abnormal cases. **Specificity** is $\mathrm{TN}/(\mathrm{TN} + \mathrm{FP})$ that measures the success of not flagging normal cases as abnormal. The ROC curve has typically horizontal axis as (1-specificity) and vertical axis as sensitivity. Once $P(\mathbf{y}_i | \mathbf{x})$ is available, the curve is generated by varying the decision threshold to discretize the probability into either 0 or 1. Despite of its clinical relevance, $P(\mathbf{y}_i | \mathbf{x})$ is intractable to compute with the model of Equ (3) due to the need of marginalizing out other binary random variables. It is however straightforward to compute with the model of Equ (1) due the independent factorization.

3. **DICE coefficient**. As a similarity measure over two sets, DICE coefficient is formulated as $\mathrm{DICE}(\mathbf{y}_\alpha, \mathbf{y}_\beta) = (2\mathbf{y}_\alpha\mathbf{y}_\beta)/(\mathbf{y}_\alpha^2 + \mathbf{y}_\beta^2) = 2\mathrm{TP}/(2\mathrm{TP} + \mathrm{FP} + \mathrm{FN})$ with the maxima at 1 when $\mathbf{y}_\alpha \equiv \mathbf{y}_\beta$. Such a metric may be generalized in cases where $\mathbf{y}_\alpha$ is a predicted probability with $\mathbf{y}_\alpha = P(\mathbf{y}|\mathbf{x})$ and $\mathbf{y}_\beta$ is the binary-valued ground truth, as is used in image

segmentation tasks such as in Ronneberger et al. (2015); Milletari et al. (2016). We adopt such a generalization as our models naturally output probabilities.

4. **Per-example sensitivity and specificity (PESS)**. The following formula is used to compute PESS

$$\text{PESS} = \frac{1}{N} \sum_{i=1}^{N} \frac{\text{sensitivity}(\hat{\mathbf{y}}_i, \mathbf{y}_i) + \text{specificity}(\hat{\mathbf{y}}_i, \mathbf{y}_i)}{2} \tag{16}$$

where $N$ is the size of the test set. Notice that the computation of sensitivity and specificity requires a binary prediction vector. Therefore, without introducing any thresholding bias, we use

$$\hat{\mathbf{y}}_i = \begin{cases} 1 & P(\mathbf{y}_i|\mathbf{x}_i) > 0.5 \\ 0 & \text{otherwise} \end{cases}$$

5. **Per-class sensitivity and specificity (PCSS)**. Unlike PESS, the following formula is used to compute PCSS

$$\text{PCSS} = \frac{1}{T} \sum_{t=1}^{T} \frac{\text{sensitivity}(\hat{\mathbf{y}}_i^t, \mathbf{y}_i^t) + \text{specificity}(\hat{\mathbf{y}}_i^t, \mathbf{y}_i^t)}{2} \tag{17}$$

where $\hat{\mathbf{y}}_i^t$ follows the same threshold of 0.5 as in PESS. Unlike PCSS where the average is over examples, PCSS averages over abnormalities instead.

## 4.3 TRAINING PROCEDURES

Three types of models are tuned on the training set. We have found that data augmentation is crucial in combatting the overfitting in all of our experiments despite their relatively small size. In particular, the input image of resolution $512 \times 512$ is randomly translated in 4 directions by 25 pixels, randomly rotated from -15 to 15 degrees, and randomly scaled between 80% and 120%. Furthermore, the ADAM optimizer Kingma & Ba (2015) is used with an initial learning rate of 0.001 which is multiplied by 0.9 whenever the performance on the validation set does not improve during training. Early stop is applied when the performance on the validation set does not improve for 10,000 parameter updates. All the reported metrics are computed on the test set with models selected with the metric in question on the validation set.

In order to ensure a fair comparison, we constrain all models to have roughly the same number of parameters. For model$_a$, where labels are considered independent, a much higher network growth rate is used for the encoder. For model$_{b_1}$ and model$_{b_2}$ where LSTMs are used as decoders, the encoders are narrower. The exact configuration of three models is shown in Table 1. In addition, we investigate the effect of ordering in the factorization of Equ (3). In particular, model$_{b_1}$ sorts labels by their frequencies in the training set while model$_{b_1}$ orders them alphabetically. All models are trained with MLE with the weighted cross-entropy loss introduced in Wang et al. (2017). All models are trained end-to-end from scratch, without any pre-training on ImageNet data.

Table 1: Hyper-parameter configuration of three models. To ensure the fairness of the comparison, we deliberately reduce the capacity of the encoder for model$_{b_1}$ and model$_{b_2}$ to match the total number of parameters of model$_a$.

|  | # of dense block $\times$ # of conv block | growth rate | LSTM dim. | total # of params |
|---|---|---|---|---|
| model$_a$ | $4 \times 3$ | 38 | - | 1,007K |
| model$_{b_1, b_2}$ | $4 \times 3$ | 19 | 100 | 1,016K |

## 4.4 QUANTITATIVE RESULTS

The AUC per abnormality is shown in Table 2, computed based on the marginal distribution of $P(\mathbf{y}|\mathbf{x})$. Only model$_a$ is included as such marginals are in general intractable for the other two due to the dependencies among $\mathbf{y}^t$s. In addition, Table 3 compares all three models based on the

proposed metrics from Section 4.2. It can be observed that our baseline model significantly outperformed the previous state-of-the-art. According to Table 3, considering label dependencies brings significant benefits in all 4 metrics and the impact of ordering seems to be marginal when the model is sufficiently trained.

Table 2: Fifteen abnormalities and their AUCs, including the average AUC over all abnormalities. The model is trained without pre-training or feature extraction from ImageNet. The model corresponds to the one in Section 3.2 where $\mathbf{y}^t$s are considered independent. This table excludes the model from Section 3.3 because AUC requires $P(\mathbf{y}^t|\mathbf{x})$, which is in general intractable.

| abnormality | Wang et al. (2017) | $model_a$ |
|---|---|---|
| atelectasis | 0.716 | **0.772** |
| cardiomegaly | 0.807 | **0.904** |
| effusion | 0.784 | **0.859** |
| infiltration | 0.609 | **0.695** |
| mass | 0.706 | **0.792** |
| nodule | 0.671 | **0.717** |
| pneumonia | 0.633 | **0.713** |
| pneumothorax | 0.806 | **0.841** |
| consolidation | 0.708 | **0.788** |
| edema | 0.835 | **0.882** |
| emphysema | 0.815 | **0.829** |
| fibrosis | 0.769 | 0.767 |
| PT | 0.708 | **0.765** |
| hernia | 0.767 | **0.914** |
| A.V.G. | 0.738 | **0.798** |
| no finding | - | 0.762 |

Table 3: Test set performance on negative log-probability (NLL), DICE, per-example sensitivity (PESS) at a threshold 0.5 and per-class sensitivity and specificity (PCSS) at a threshold of 0.5. See Section 4.2 for explanations of the metrics. In addition to $model_a$ used in Table 2, $model_{b1}$ and $model_{b2}$ corresponds to the model introduced in Section 3.3, with the difference in the ordering of the factorization in Equ (3). $model_{b1}$ sorts labels by their frequency in the training set in ascending order. As a comparison, $model_{b2}$ orders labels alphabetically according to the name of the abnormality.

| | NLL | DICE | $PESS_{0.5}$ | $PCSS_{0.5}$ |
|---|---|---|---|---|
| $model_a$ | 4.474 | 0.261 | 0.752 | 0.665 |
| $model_{b1}$ | 4.099 | **0.310** | **0.765** | **0.676** |
| $model_{b2}$ | **3.848** | **0.310** | **0.767** | **0.677** |

## 5  CONCLUSION

To improve the quality of computer-assisted diagnosis of chest x-rays, we proposed a two-stage end-to-end neural network model that combines a densely connected image encoder with a recurrent neural network decoder. The first stage was chosen to address the challenges to learning presented by high-resolution medical images and limited training set sizes. The second stage was designed to allow the model to exploit statistical dependencies between labels in order to improve the accuracy of its predictions. Finally, the model was trained from scratch to ensure that the best application-specific features were captured. Our experiments have demonstrated both the feasibility and effectiveness of this approach. Indeed, our baseline model significantly outperformed the current state-of-the-art. The proposed set of metrics provides a meaningful quantification of this performance and will facilitate comparisons with future work.

While a limited exploration into the value of learning interdependencies among labels yields promising results, additional experimentation will be required to further explore the potential of this

methodology both as it applies specifically to chest x-rays and to medical diagnostics as a whole. One potential concern with this approach is the risk of learning biased interdependencies from a limited training set which does not accurately represent a realistic distribution of pathologies – if every example of cardiomegaly is also one of cardiac failure, the model may learn to depend too much on the presence of other patterns such as edemas which do not always accompany enlargement of the cardiac silhouette. This risk is heightened when dealing with data labeled with a scheme which mixes pathologies, such as pneumonia, with patterns symptomatic of those pathologies, such as consolidation. The best approach to maximizing feature extraction and leveraging interdependencies among target labels likely entails training from data labeled with an ontology that inherently poses some consistent known relational structure. This will be the endpoint of a future study.

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
