# OpenReview forum: "Learning to diagnose from scratch by exploiting dependencies among labels"
_ICLR.cc/2018/Conference — Reject_

### Official Review · AnonReviewer3 · 2017-11-28
**Learning to diagnose from scratch by exploiting dependencies among labels**

**Rating:** 6
**Confidence:** 4

**Review:**

This paper presents an impressive set of results on predicting lung pathologies from chest x-ray images.
Authors present two architectures: one based on denseNet, and one based on denseNet + LSTM on output dimensions (i.e. similar to NADE model), and compare it to state of the art on the chest x-ray classification. Experiments are clearly described and results are significantly better compared to state of the art.

The only issue with this paper is, that their proposed method, in practice is not tractable for inference on estimating probability of a single output, a task which would be critical in medical domain. Considering that their paper is titled as a work to use "dependencies" among labels, not being able to evaluate their network's, and lack of interpretable evaluation results on this model in the experiment section is a major limitation.

On the other hand, there are many alternative models where one could simply use multi-task learning and shared parameter, to predict multiple outcomes extremely efficiently. To be able to claim that this paper improved the prediction by better modeling of 'dependencies' among labels, I would need to see how the (much simpler) multi-task setting works as well.

That said, the paper has several positive aspects in all areas:

Originality - the paper presents first combination of DenseNets with LSTM-based output factorization,
Writing clarity - the paper is very well written and clear.
Quality - (apart from the missing multi-task baseline), the results are significantly better than state of the art, and experiments are well done,
Significance - Apart from the issue of intractable inference which is arguably a large limitation of this work, the application in medical field is significant.

---

> ### Public Comment · (anonymous) · 2017-12-24
> **Author Rebuttal**
>
> "The only issue with this paper is, that their proposed method, in practice is not tractable for inference on estimating probability of a single output, a task which would be critical in medical domain. Considering that their paper is titled as a work to use "dependencies" among labels, not being able to evaluate their network's, and lack of interpretable evaluation results on this model in the experiment section is a major limitation."
>
> The reviewer is correct that estimating the marginal probability of a given abnormality using the proposed model would be computationally expensive. We do not consider this a major limitation because the motivation for our research was to model events (i.e. the abnormalities) that occur together. By definition, the marginal probability describes each event in isolation. Section 1 of the paper provides a few specific examples of abnormalities that domain experts know to be dependent. The proposed model attempts to capture such dependencies by describing the joint distribution abnormalities. The medical use case that we intended to support is for the model to predict all abnormalities that are present in the image and the joint probability quantifies the confidence of the prediction as a whole. That being said, other metrics such as sensitivities and specificities remain accessible (Section 4.2) for individual abnormalities.
>
> "On the other hand, there are many alternative models where one could simply use multi-task learning and shared parameter, to predict multiple outcomes extremely efficiently. To be able to claim that this paper improved the prediction by better modeling of 'dependencies' among labels, I would need to see how the (much simpler) multi-task setting works as well."
>
> The baseline model (model_{a}) described in our paper represents the standard multi-task learning approach in which the encoder parameters are shared across classes and decoder uses class-specific output layers. The alternative models (model_{b1} and model_{b2}) employ a comparable encoder architecture but a recurrent decoder. Our claim that modeling dependencies improved the predictions is based entirely on the a comparison you have suggested.

---

### Official Review · AnonReviewer1 · 2017-12-02
**Interesting but flawed in terms of applicability**

**Rating:** 6
**Confidence:** 3

**Review:**

Well written and appropriately structured. Well within the remit of the conference.
Not much technical novelty to be found, but the original contributions are adequately identified and they are interesting on their own.

My main concern (and complaint) is not technical, but application-based. This study is (unfortunately) typical in that it focuses on and provides detail of the technical modeling issues, but ignores the medical applicability of the model and results. This is exemplified by the fact that the data set is hardly described at all and the 14 abnormalities/pathologies, the rationale behind their choice and the possible interrelations and dependencies are never described from a medical viewpoint. If I were a medical expert, I would not have a clue about how these results and models could be applied in practice, or about what medical insight I could achieve.

The bottom line seems to be: "my model and approach works better than the other guys' model and approach", but one is left with the impression that these experiments could have been made with other data, other problems, other fields of application and they would not have not changed much

---

> ### Public Comment · (anonymous) · 2017-12-24
> **Author Rebuttal**
>
> Thank you for taking time reviewing our manuscript. Much effort has been devoted into making this paper digestible for both machine learning researchers and medical practitioners. We are surprised that you have found none of them useful. For instance, the rationale behind our modelling choices is carefully phrased from the medical point of view in length in Section 1 (especially paragraph 2), Section 3.1, Section 3.3.1, Section 4.2 and Section 5. In fact, wherever relevant, we have made an effort to motivate and justify the modeling decisions with the medical application in question, although there might still be places where the narrative could be improved to draw a deeper connection. Regarding your point of the dataset, Wang et al. (2017) (the paper introducing the dataset to the community) has thorough descriptions of its curation, context, label distribution. That being said, we agree with you that this is not strictly speaking a medicine-oriented publication that is meant to guide the clinical practice, as rigorous work in that category would require conducting a randomized clinical trial in the hospital environment to systematically measure its clinical outcome.

---

### Official Review · AnonReviewer2 · 2017-12-13
**Interesting application of DenseNet/LSTM to the medical field**

**Rating:** 6
**Confidence:** 3

**Review:**

The paper proposes to combine the recently proposed DenseNet architecture with LSTMs to tackle the problem of predicting different pathologic patterns from chest x-rays. In particular, the use of LSTMs helps take into account interdependencies between pattern labels.

Strengths:
- The paper is very well written. Contextualization with respect to previous work is adequate. Explanations are clear. Novelties are clearly identified by the authors.
- Quantitative improvement with respect to the state the art.

Weaknesses:
- The paper does not introduce strong technical novelties -- mostly, it seems to apply previous techniques to the medical domain. It could have been interesting to know if there are more insights / lessons learned in this process. This could be of interest for a broader audience. For instance, what are the implications of using higher-resolution images as input to DenseNet / decreasing the number of layers? How do the features learned at different layers compare to the ones of the original network trained for image classification? How do features of networks pre-trained on ImageNet, and then fine-tuned for the medical domain, compare to features learned from medical images from scratch?
- The impact of the proposed approach on medical diagnostics is unclear. The authors could better discuss how the approach could be adopted in practice. Also, it could be interesting also to discuss how the results in Table 2 and 3 compare to human classification capabilities, and if that performance would be already enough for building a computer-aided diagnosis system.

Finally -- is it expected that the ordering of the factorization in Eq. 3 does not count much (results in Table 3)? As a non-expert in the field, I'd expect that ordering between pathologic patterns matters more.

---

> ### Public Comment · (anonymous) · 2017-12-24
> **Author Rebuttal**
>
> "The paper does not introduce strong technical novelties -- mostly, it seems to apply previous techniques to the medical domain."
>
> The NADE-LSTM hybrid model has not previously been explored. Leveraging the label dependency has been ignored in previous work on medical diagnosis. We carefully designed such models to address it while taking into account application-specific constraints (e.g.  Section 1 (especially paragraph 2), Section 3.1, Section 3.3.1 and Section 5). In addition, clinical-relevant metrics are proposed, analyzed (see Section 4.2) and measured (see Section 4.4) as opposed to conventional machine learning metrics that are hard to interpret clinically by medical practitioners. We believe all of the above contributions are novel.
>
> "It could have been interesting to know if there are more insights / lessons learned in this process. This could be of interest for a broader audience. For instance, what are the implications of using higher-resolution images as input to DenseNet / decreasing the number of layers? How do the features learned at different layers compare to the ones of the original network trained for image classification? How do features of networks pre-trained on ImageNet, and then fine-tuned for the medical domain, compare to features learned from medical images from scratch?"
>
> Thank you for the suggestions. The central idea is to exploit the label dependencies, for which we have focus on extensively. For other non-central design choices (such as architectural choice of DenseNets, the use of fine-tuning), the reasoning has been stated while leaving quantitative evaluation for the future study. Regarding your point of pre-trained models: Table 2 shows the diminishing benefits of using pretrained out-of-domain models once a large amount of in-domain training pairs are available, as the previous SOTA relies a pretrained model from ImageNet. For both pretraining and fine-tuning, lower resolution (224*224, as opposed to 512*512 in our work) and out-of-domain bias (from ImageNet, very much different from medical images) could possibly account for the difference. As you mentioned, fine-tuning might be a reasonable middle-ground. However, as has been recently observed, fine-tuning has its inherent issue of “catastrophic forgetting” and needs to be dealt with care.
>
> "The impact of the proposed approach on medical diagnostics is unclear. The authors could better discuss how the approach could be adopted in practice. Also, it could be interesting also to discuss how the results in Table 2 and 3 compare to human classification capabilities, and if that performance would be already enough for building a computer-aided diagnosis system."
>
> The medical use case that motivated our research was the automated prediction of all (modeled) abnormalities that are present in a chest x-ray image. While the performance of the proposed model represented an improvement over the previous state of the art, the accuracy certainly falls short of human performance. In practice, however, the predictions are probably good enough to find utility in triage applications or as a second-read diagnostic aid. Measurement of the clinical impact, as well as a comparison to human performance, will have to wait for future studies designed to collect and analyze the required data in a standard randomized clinical trial.
>
> "Finally -- is it expected that the ordering of the factorization in Eq. 3 does not count much (results in Table 3)? As a non-expert in the field, I'd expect that ordering between pathologic patterns matters more."
>
> Just to be clear, this is an empirical question. In theory, all orderings are equivalent but in practice they might differ because factors are parameterized by models. Based on our experiments, however, the ordering of the factorization does not significantly impact the results. This phenomenon has been consistently observed in other publications that use NADEs (e.g. Iterative Neural Autoregressive Distribution Estimator, NIPS 2014). In practice, one could average models trained with different orderings. Model averaging is not the central point of this work.

---

### Decision · Program_Chairs · 2018-01-29
**ICLR 2018 Conference Acceptance Decision**

**Decision:**

Reject

**Comment:**

Authors apply dense nets and LSTM to model dependencies among labels and demonstrate new state-of-art performance on an X-Ray dataset.

Pros:
- Well written.
- New improvement to state-of-art

Cons:
- Novelties are not strong. One combination of existing approaches are used to achieve state-of-art on what is still a relatively new dataset. (All Reviewers)

- Using LSTM to model dependencies would be affected by the selected order of the disease states. In this sense, LSTM seems like the wrong architecture to use to model dependencies among labels. This may be a drawback in comparison to other methods of modeling dependencies, but this is not thoroughly discussed or evaluated. (Reviewer 1 & 3)

- There is a large body of work on multi-task learning with shared information, which have not been evaluated for comparison. Because of this, the contribution of the LSTM to model dependencies between labels in comparison to other available approaches cannot be verified. (Reviewer 1 & 3)

- Top AUC performance on this dataset does not carry much significance on its own, as the dataset is new (CVPR 2017), and few approaches have been tested against it.

- Medical literature not cited to justify with evidence the discovered dependencies among disease states. (Reviewer 1)